# Hippocampal Atrophy in Pediatric Transplant Recipients with Human Herpesvirus 6B

**DOI:** 10.3390/microorganisms9040776

**Published:** 2021-04-08

**Authors:** Misa Miyake, Yoshiki Kawamura, Naoko Ishihara, Shigetaka Suzuki, Hiroki Miura, Yoko Sakaguchi, Masaharu Tanaka, Yoshiyuki Takahashi, Seiji Kojima, Hiroshi Toyama, Jun Natsume, Tetsushi Yoshikawa

**Affiliations:** 1Department of Pediatrics, Fujita Health University School of Medicine, Toyoake 470-1192, Japan; mh0127@fujita-hu.ac.jp (M.M.); naokois@fujita-hu.ac.jp (N.I.); hiroki-m@fujita-hu.ac.jp (H.M.); tetsushi@fujita-hu.ac.jp (T.Y.); 2Department of Radiology, Fujita Health University School of Medicine, Toyoake 470-1192, Japan; fish.shigetakai@gmail.com (S.S.); masaharutan32@gmail.com (M.T.); 3Department of Pediatrics, Nagoya University Graduate School of Medicine, Nagoya 466-8560, Japan; yoko.s@med.nagoya-u.ac.jp (Y.S.); ytakaha@med.nagoya-u.ac.jp (Y.T.); kojimas@med.nagoya-u.ac.jp (S.K.); htoyama@fujita-hu.ac.jp (H.T.); junnatsu@med.nagoya-u.ac.jp (J.N.)

**Keywords:** human herpesvirus 6B, hippocampus, children, hematopoietic stem cell transplant

## Abstract

The aim of this study was to determine whether human herpesvirus 6B (HHV-6B) infection can impair the hippocampus in pediatric hematopoietic stem cell transplant (HSCT) recipients. Study subjects were pediatric HSCT recipients monitored for HHV-6B infection who underwent brain MRI before and after transplantation. Volumetric analysis of the hippocampus was performed. Of the 107 patients that received HSCT at Nagoya University Hospital Between July 2008 and April 2014, 20 were eligible for volumetric analysis. Eight patients had HHV-6B infection, of whom two had encephalopathy at the time of HHV-6B infection. None of the 12 patients without HHV-6B infection had encephalopathy. The median ratio of the right hippocampal volume from before to after transplantation was 0.93 in patients with HHV-6B infection and 1.02 in without HHV-6B infection (*p* = 0.007). The median ratio of the left hippocampal volume ratio in patients with and without HHV-6B infection was 0.92 and 1.00, respectively (*p* = 0.003). Among the eight patients with HHV-6B infection, four had a marked reduction in hippocampal volume (volume ratio < 0.90). Only one of these patients had neurological symptoms at the time of HHV-6B infection. The reduction in the hippocampal volume ratio was higher in pediatric HSCT recipients with HHV-6B infection than those without viral infection. Neurological follow-up may be required for pediatric HSCT recipients with HHV-6B infection.

## 1. Introduction

Primary human herpesvirus 6B (HHV-6B) infection can cause exanthem subitum [1]. Although the disease is a benign febrile illness in young children, central nervous system (CNS) complications such as febrile seizures [2,3,4] and encephalitis [5,6,7,8] can sometimes occur. After the primary infection, HHV-6B establishes latency in various body sites, including the CNS [9,10,11]. HHV-6B can reactivate in immunocompromised patients, such as hematopoietic stem cell transplantation (HSCT) recipients [12,13,14,15,16]. Viral reactivation may be associated with a variety of diseases after transplantation [12,15,17,18,19,20]. In particular, HHV-6B is a major cause of post-transplant acute limbic encephalitis (PALE) [21,22,23,24]. Furthermore, it has been suggested that HHV-6B is involved in the pathogenesis of temporal lobe epilepsy, a type of refractory adult epilepsy [25,26].

Typical clinical symptoms of PALE include delirium and memory loss. Typical brain MRI findings include hyperintensities in the medial temporal lobes, including the hippocampus. It has been demonstrated that cerebrospinal fluid (CSF) viral DNA loads are very low or undetectable in children with HHV-6B encephalitis at the time of primary viral infection [27]. However, high amounts of HHV-6B DNA are detected in the CSF of adult PALE patients [24,27], which decrease quickly with antiviral drug administration [28]. These data suggest that HHV-6B replication in the hippocampus may cause direct neurological impairment in patients with PALE.

Most previously reported patients with PALE were adult transplant recipients [22,29,30]. By contrast, a limited number of pediatric patients with PALE have been reported to date [31]. Our recent retrospective study showed a low prevalence of HHV-6B encephalitis in pediatric HSCT recipients [32]. It has been suggested that HHV-6B infection is associated with mild cognitive impairment in pediatric HSCT recipients [33]. Therefore, the aim of this study was to determine whether HHV-6B infection can impair the hippocampus in pediatric HSCT transplant recipients based on volumetric analysis of the hippocampus before and after HSCT with brain MRI.

## 2. Materials and Methods

### 2.1. Patients

Between July 2008 and April 2014, 107 patients received HSCT at Nagoya University Hospital. They were monitored weekly for HHV-6B infection with viral isolation and real-time PCR. Among these patients, 63 were excluded because of missing pre- or post-transplant brain MRI. An additional nine patients were excluded because they had CNS disorders such as brain tumors. Thus, 35 of the 107 HSCT recipients were monitored for HHV-6B infection and underwent brain MRI before and after transplantation. However, 15 patients were excluded due to inadequate brain MR images (unclear image, two patients; different slice thickness before and after transplantation, 13 patients). Ultimately, 20 patients were enrolled in the volumetric analysis (Figure 1). Clinical information was collected retrospectively from medical records.

### 2.2. Sample Collection and Definition of HHV-6B Infection

Ethylenediaminetetraacetic acid-treated peripheral blood samples were collected weekly from HSCT recipients to monitor for HHV-6B infection. Peripheral blood mononuclear cells (PBMCs) were separated for viral isolation. DNA was extracted from 200 µL of whole blood using the QIAamp DNA Blood Mini Kit (QIAGEN, Hilden, Germany). Real-time PCR was carried out to measure HHV-6B DNA load in whole blood samples. Patients were identified as having HHV-6B infection if they had a positive viral isolation or detection of more than 1 × 10^4^ copies/mL of viral DNA in whole blood.

### 2.3. HHV-6B Isolation

The method for HHV-6 isolation was previously described [34]. PBMCs were co-cultured with pre-stimulated cord blood mononuclear cells. Positive cultures were confirmed by indirect immunofluorescence staining with an HHV-6B specific monoclonal antibody (OHV-3).

### 2.4. Real-Time Quantitative PCR Assay for HHV-6

A PCR probe and primers for HHV-6B were selected from the U31 gene (large tegument protein). The upstream primer was 5′-TTTGCAGTCATCACGATCGG-3′ and the downstream primer was 5′-AGAGCGACAAATTGGAGGTTTC-3′. A fluorogenic probe (5′-AGCCACAGCAGCCATCTACATCTGTCAA-3′) was located between the primers. The PCR reaction was performed using a TaqMan PCR kit (PE Applied Biosystems, Foster City, CA, USA) as previously described [35]. In samples positive for HHV-6B DNA, the HHV-6B species was determined based on restriction fragment length polymorphism analysis of loop-mediated isothermal amplification products as previously described [36]. It was confirmed that HHV-6 DNA was not detected by real-time PCR in the whole blood cells of recipients before transplantation.

### 2.5. MRI

Brain MRI was performed before and after transplantation to evaluate neurological complications caused by chemotherapy and transplantation conditioning treatments. Four different MRI scanners were used: MAGNETOM Trio (Siemens, Erlangen, Germany), MAGNETOM Verio (Siemens), MAGNETOM Aera (Siemens), and Atlas SPEEDER (Canon Medical Systems, Otawara, Japan). The MRI scanning conditions are shown in Appendix A.

### 2.6. Hippocampal Volumetric Analysis

Volumetric analysis was performed with Workstation (AZE Virtual Place Raijin, AZE, Kawasaki, Japan). Hippocampal delineation was performed manually in coronal T2-weighted slices. Since the hippocampus is divided into three parts (head, body, and tail), we manually measured the three sections surrounding the hippocampus from head to tail. A representative image is shown in Figure 2. Volumes were automatically calculated from the surrounding area and slice thickness. Consequently, patients with different slice thicknesses as evidenced by the MRI examinations before and after HSCT were excluded from the analysis. The hippocampal volume ratio was calculated by dividing the volume after transplantation by the volume before transplantation. In order to eliminate bias in volume evaluation, M.M. measured the volumes for each patient while blinded to HHV-6B infection status.

### 2.7. Statistical Analysis

Clinical characteristics such as gender, underlying disease, transplant type, donor allotype, total body irradiation, and MRI slice thickness were compared between patients with and without HHV-6B infection using Fisher’s exact test. Patient age, number of days of receiving MRI, and hippocampal volume ratio were also compared between patients with and without HHV-6B infection using the Mann–Whitney U-test. A statistically significant difference was defined as *p* < 0.05. Statistical analysis was performed using JMP version 13.2.1 (SAS Institute, Cary, NC, USA).

### 2.8. Ethical Approval

This study was approved by the Ethics Review Board for Human Studies at Fujita Health University (Accession number HM-17-484, approved on 13 March 2018). Informed consent was obtained in the form of opt-out. Those who declined to participate were excluded from the study.

## 3. Results

HHV-6B infection was demonstrated in 8 of the 20 patients after HSCT. At the time of viral infection, two of the eight patients with HHV-6B infection had encephalopathy. Encephalopathy was not observed in the 12 patients without HHV-6B infection. The patient characteristics and demographic factors are summarized in Table 1. The numbers of male and female patients without HHV-6B infection were six and six, respectively, but seven of the eight patients with HHV-6B infection were male (*p* = 0.158). All other variables, including median age (*p* = 0.416), underlying disease (*p* = 0.112), graft type (*p* = 0.450), donor type (*p* = 1.000), total body irradiation (*p* = 1.000), and timing of pre- (*p* = 0.129) and post-transplant brain MRI (*p* = 0.355) were not significantly different between patients with and without HHV-6B infection.

The hippocampal volume ratio before and after transplantation was compared between recipients with and without HHV-6B infection (Figure 3). In the right hippocampus, the median volume ratio was 0.93 (range, 0.83–1.02) in patients with HHV-6B infection and 1.02 (range, 0.97–1.03) in patients without HHV-6B infection (*p* = 0.007). In the left hippocampus, the median volume ratio in patients with and without HHV-6B infection was 0.92 (range, 0.86–1.00) and 1.00 (range, 0.98–1.03) (*p* = 0.003), respectively.

The detailed clinical, virological, and radiological findings are summarized in Table 1. Among the eight patients with HHV-6B infection, Patients 1–4 had remarkably low hippocampal volume ratios (Table 2). The brain MRI of Patient 1 is shown as a representative MR image (Figure 2). Among the four patients, the highest copy numbers of HHV-6B DNA in the peripheral blood occurred two to six weeks after transplant. HHV-6B was isolated only from Patient 1 at 18 days after transplant. Although only Patient 3 had neurological symptoms such as seizures and was diagnosed with posterior reversible encephalopathy syndrome based on the presence of hypertension and typical brain MRI findings, such as hyperintensities in the occipital lobe on the T2-weighted image, no neurological symptoms were observed in the other three patients (Table 2). In Patient 3, no viral DNA was detected in the CSF. As neurological symptoms improved with administration of antihypertensive medication, no antiviral treatment was carried out in this patient. The remaining four patients (Patients 5–8) did not have any remarkable reductions in hippocampal volume. Although Patient 8 did not have hippocampal atrophy, he had posterior reversible encephalopathy syndrome at the time of HHV-6B infection and large amounts of HHV-6B DNA were detected in the CSF [37]. This patient received antiviral therapy with a decrease in viral DNA load. As shown in Table 2, the interval between HHV-6B infection and post-transplant MRI examination was short in two patients; it was 10 days in Patient 7 and nine days in Patient 8.

## 4. Discussion

Hippocampal MRI volumetric analysis is used to elucidate the pathophysiology of various types of neurological diseases, such as mesial temporal lobe epilepsy, Alzheimer’s disease, and post-traumatic stress disorder [38,39,40]. Volumetric analysis based on neuroimaging examinations provides clear evidence of hippocampal damage, even when clinical symptoms are mild or ambiguous. As described above, our recent retrospective study demonstrated that HHV-6B-associated CNS disease in pediatric HSCT recipients occurs less frequently than that in adult HSCT recipients. The clinical symptoms also differ between pediatric and adult HSCT recipients [32]. It is possible that mild cognitive impairment caused by HHV-6B infection may be missed in pediatric patients. Therefore, we thought that the hippocampal MRI volumetric analysis might be useful for assessing the neurological impairment caused by HHV-6B infection in pediatric HSCT recipients.

Although the number of patients was limited, a statistically significant reduction in bilateral hippocampal volume was demonstrated in patients with HHV-6B infection (right: *p* = 0.007; left: *p* = 0.003). Post-transplant brain MRI was carried out approximately 80 days after transplant in this cohort. HHV-6B infection occurred approximately two to three weeks after transplant (median: 18 days after transplant; range: 12–47 days after transplant). Post-transplant MRI was performed approximately two months after HHV-6B infection. It has previously been demonstrated that typical brain MRI findings such as abnormal signal intensity in the mesial temporal lobe are generally observed in the acute phase of PALE [22,28,41]. Meanwhile, hippocampal atrophy was demonstrated in the recovery phase (two years after the onset of PALE) in a pediatric patient with HHV-6B-associated PALE [31]. Only two of the eight patients with HHV-6B infection had neurological symptoms and were diagnosed with posterior reversible encephalopathy syndrome. The remaining six patients may have had no symptoms or mild neurological symptoms that were difficult to detect. Therefore, the present study and previous studies suggest that HHV-6B infection may cause hippocampal atrophy in pediatric HSCT recipients without the typical clinical features of PALE.

Since a remarkable reduction in hippocampal volume was demonstrated in four of the eight patients with HHV-6B infection (Figure 3 and Table 2), the background and clinical information of the eight patients with HHV-6B infection are summarized in Table 2 to identify the factors associated with the marked reduction. HHV-6B infection occurred approximately two to six weeks after transplant based on the real-time PCR analysis and viral isolation. Although all four patients with a remarkable reduction underwent MRI long after HHV-6B infection, two of the four patients without a remarkable reduction underwent MRI soon after HHV-6B infection (10 days in Patient 7 and nine days in Patient 8), which might be too early to detect hippocampal atrophy. Since Patient 8 had HHV-6B-associated posterior reversible encephalopathy syndrome with large amounts of viral DNA in the CSF [37], we thought that follow-up MRI examination was necessary during the recovery phase.

It is difficult to understand why no remarkable neurological signs or symptoms were observed in six patients, even though three had a remarkable reduction in hippocampal volume. Three previously reported cases of pediatric PALE involved confusion and behavioral changes approximately 20–23 days after HSCT, and two of the three patients also had seizures [42]. Since the typical clinical manifestations of PALE, such as behavioral changes and short-term memory loss, might be difficult to detect in pediatric patients, some patients in this age group with PALE might not have been diagnosed precisely. When Zerr et al. prospectively evaluated 315 HSCT recipients to investigate the association between HHV-6B infection and delirium or cognitive dysfunction, they found that patients with HHV-6B infection are more likely to develop delirium and have neurocognitive dysfunction [33]. They used a modified delirium rating scale and neuropsychological testing specific for pediatric patients. Therefore, in order to evaluate the clinical features of HHV-6B-associated CNS disease in pediatric HSCT recipients, sensitive diagnostic methods to identify mild neurological symptoms should be used at the time of HHV-6B infection. In addition, long-term neurological and neuroimaging follow-up of pediatric HSCT recipients with HHV-6B infection is needed.

As a limitation, HHV-6B infection could not be identified in terms of whether the infection was from donors or from the reactivation in recipients in this study, because the HHV-6 antibodies in recipients and the viral DNA in donors before transplantation were not measured. Since this is important in considering the effect on recipients, it is necessary to identify infection from donors or reactivation in recipients in future studies. Other study limitations include the small number of cases and the manual assessment of hippocampal volume using the AZE system. Since brain MRI was performed to screen for CNS complications induced by chemotherapy in this study, imaging conditions and timing of examinations varied by patient condition. Furthermore, only two-dimensional images were used during the study period. On the contrary, automatic volumetric analysis was recently used to measure hippocampal volume precisely using the latest three-dimensional image analysis techniques such as FreeSurfer (https://surfer.nmr.mgh.harvard.edu/, accessed on 7 August 2020). Furthermore, the hippocampus has been recognized as a structure consisting of several subfields with distinct histological characteristics [43]. An association between subtle pathophysiological changes in the hippocampus and mild neurological symptoms has been discovered in patients with Parkinson’s disease [44] and mesial temporal lobe sclerosis [45] with the latest three-dimensional image analysis techniques. Therefore, evaluation of hippocampal subfields using the same analysis techniques may provide additional information to clarify HHV-6B-associated CNS disease in pediatric HSCT recipients. Thus, in order to support the present findings, a large number of pediatric HSCT recipients should be prospectively evaluated using the latest MRI volumetric analysis techniques in a future study.

## Figures and Tables

**Figure 1 microorganisms-09-00776-f001:**
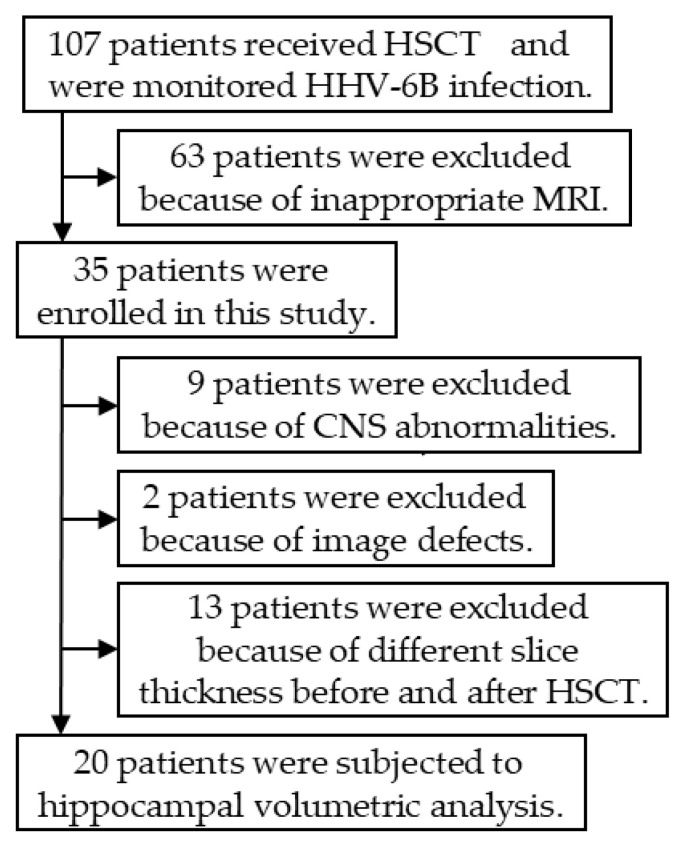
Flow chart of the inclusion and exclusion criteria for hippocampal volumetric analysis. HHV-6B, human herpesvirus 6B; HSCT, hematopoietic stem cell transplant.

**Figure 2 microorganisms-09-00776-f002:**
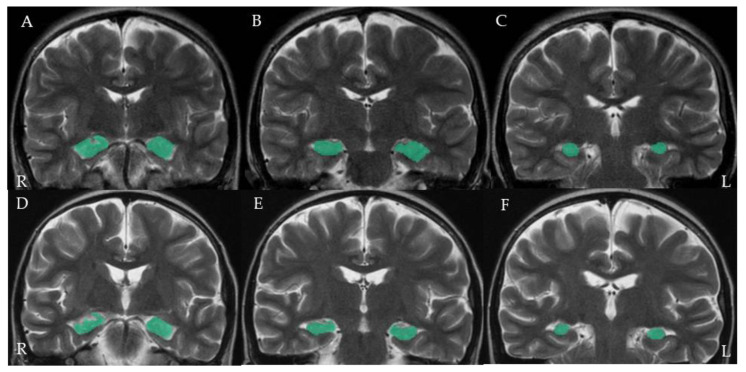
A coronal T2-weighted image of the brain for patient 1. The three hippocampal segments (head, body, and tail) before transplantation (**A**–**C**) and after transplantation (**D**–**F**) are shown in green.

**Figure 3 microorganisms-09-00776-f003:**
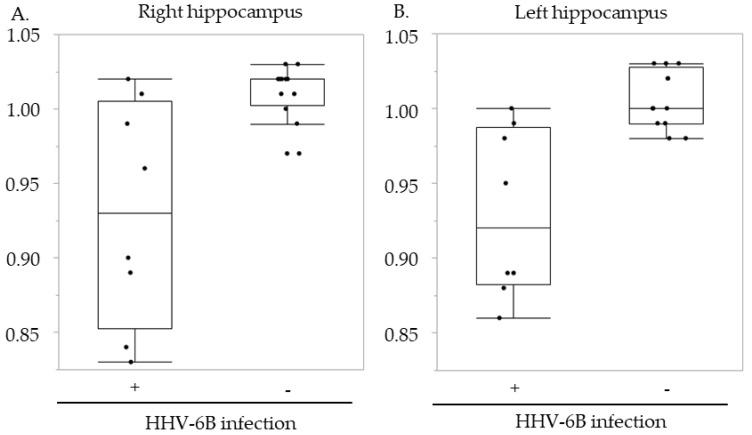
The hippocampal volume ratio before and after transplantation. (**A**) Right hippocampus; (**B**) Left hippocampus. The vertical axes indicate the hippocampal ratio of the hippocampal volume from before to after hematopoietic stem cell transplantation recipients with (+) and without (−) human herpesvirus 6B (HHV-6B) infection.

**Table 1 microorganisms-09-00776-t001:** Characteristics and clinical features of the patients.

Variable	HHV-6B Infection (*n* = 8)	No HHV-6B Infection (*n* = 12)	*p*-Value
Median age (range), years	10.5 (3–15)	8.5 (2–14)	0.416
Sex			0.158
Male	7	6	
Female	1	6	
Underlying disease			0.112
Hematologic malignancy	5	3	
Solid tumor	2	2	
Other	1	7	
Transplant type			0.450
BM	6	10	
PBSC	1	2	
CB	1	0	
Donor allotype			1.000
Related	4	7	
Unrelated	2	3	
TBI			1.000
Yes	6	10	
No	2	2	
Duration between transplant and MRI examination Median (range), days			
Before	−16 (−12 to −57)	−22 (−13 to −35)	0.129
After	74 (26 to 153)	87 (43 to 239)	0.335

Abbreviations: BM, bone marrow; CB, cord blood; HHV-6, human herpesvirus 6B; PBSC, peripheral blood stem cell; TBI, total body irradiation.

**Table 2 microorganisms-09-00776-t002:** Characteristics and laboratory and MRI findings of eight patients with HHV-6B infection.

Patient	Age Years/Sex	Underlying Disease	Transplant Type	TBI	CNS Symptoms	HHV-6B DNA in Whole Blood ^a^ (Day of Sample ^b^)	Isolation of HHV-6B (Day of Sample ^b^)	Median Duration between Transplant and MRI Examination ^b^ before/after	Hippocampal Volume Ratio	
									Right	Left
1	12/M	CML	BM	Yes	No	61,300 (18)	Yes (18)	−16/84	0.83	0.88
2	5/M	MDS	BM	Yes	No	104,600 (19)	No	−57/153	0.84	0.89
3	10/F	AML	BM	Yes	Yes	44,600 (12)	No	−12/28	0.89	0.89
4	6/M	Adrenal tumor	PBSC	No	No	12,200 (18)	No	−16/69	0.90	0.86
5	3/M	NB	CB	Yes	No	348,400 (22)	Yes (13)	−15/126	0.96	0.98
6	15/M	AA	BM	No	No	Not detected	Yes (26)	−16/79	0.99	0.95
7	15/M	CML	BM	Yes	No	85,100 (47)	Yes (47)	−16/57	1.01	0.99
8	11/M	ALL	BM	Yes	Yes	14,462,400 (17)	No	−21/26	1.02	1.00

Abbreviations: AA, anaplastic anemia; ALL, acute lymphocytic leukemia; AML, acute myeloid leukemia; BM, bone marrow; CML, chronic myeloid leukemia; CNS, central nervous system; HHV-6B, human herpesvirus 6B; MDS, myelodysplastic syndrome; NB, neuroblastoma; PBSC, peripheral blood stem cell; TBI, total body irradiation. ^a^ Copies/mL; ^b^ day of transplantation defined as day 0.

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
