# Peer review of "Hippocampal Atrophy in Pediatric Transplant Recipients with Human Herpesvirus 6B"

_microorganisms, 2021, doi:10.3390/microorganisms9040776_

Round 1

Reviewer 1 Report

Miyake et al., performed a study analyzing influence of HHV-6B infection on hippocampus impairment in pediatric HSCT patients. They find that the reduction in hippocampal volume ratio was higher in patients positive for HHV6B.

The manuscript addresses an important and not broadly studied topic and therefore as such is of interest to the readers. It is quite remarkable that the authors observed such a clear effect of HHV6B presence on hippocampal atrophy, considering low number of patients involved in the final study.

Comments:

  1. several points in the introduction should be clarified and described in more detail: line 61 states: "It has been demonstrated that CSF viral DNA loads are very low or undetectable in patients with HHV-6B encephalitis at the time of primary viral infection." Does this statement refer to adult, pediatric or both types of patients? Would it be possible to put a reference for the source of this information? Do the authors really mean a primary HHV6B infection here or rather a reactivation, as stated in the first paragraph of the Introduction? Is it possible to differentiate the mode of infection (primary vs reactivation) in this clinical context? What evidence do the authors have for the statement that it is a HHV6B infection rather than reactivation - is there data available to show that patients and transplants were virus negative before the transplant? Please state what other diseases of the CNS can be caused by HHV-6B reactivation in transplant patients and why this study focuses on PALE.
  2. In the Results section please clarify the following: please specify what criteria determined whether viral DNA isolation was performed or not and on which day  it was performed? Does "No" in the column "Isolation of HHV-6B" mean the isolation was not performed or isolation was not successful? Were the patients tested for the HHV-6B before the transplant? Were the transplant donors tested for HHV-6B? Would it be possible to perform some of these tests now ie. are the samples from before transplantation still available? In table 2 for patient #6 it is stated that the HHV-6B was not detected - on which day was the test performed?
  3. In the "Discussion" please clarify the following: Are there treatments available for the HHV-6B-associated reactivation in HSCT pediatric patients (or adult patients)? If so, what are they and are they successful? Do the results of this study suggest that detection of hippocampal atrophy could be used as an early predictive marker for PALE in pediatric patients or do the authors suspect that the classical PALE symptoms will not appear in these patients?How long after the transplant were the patients followed up for symptoms of PALE.

Minor comments: in lines 188 and 265 the numbers representing references are not in brackets.

Reviewer 2 Report

In this manuscript, Miyake et al. investigated the potential consequences of HHV-6B infection on the hippocampus of pediatric patients receiving hematopoietic stem cell patients (HSCT).  MRI and volumetric analysis before and after HSCT was performed.  Twelve of twenty patients did not have HHV-6B patients and none of these patients had encephalopathy or a reduction in hippocampal volume.  In contrast, four of the eight patients with HHV-6B infection had a reduction in hippocampal volume; only one of these had neurologic symptoms at the time of infection.  This study suggests a correlation between HHV-6 B infection and hippocampal volume loss and has potential implications for understanding the underlying neuropathology of post-transplant acute limbic encephalitis (PALE) and temporal lobe epilepsy, which have both been associated with HHV-6B infection.

Minor concern

HHV-6A is also highly neurotropic and has also been associated with neurologic disease, including MS and Alzheimer’s.  It would be of interest to also determine whether HHV-6A DNA was present in the PBMC of the twenty patients where volumetric analysis was possible.

Author Response

This manuscript is a resubmission of an earlier submission. The following is a list of the peer review reports and author responses from that submission.